# The Roles of the IGF Axis in the Regulation of the Metabolism: Interaction and Difference between Insulin Receptor Signaling and IGF-I Receptor Signaling

**DOI:** 10.3390/ijms22136817

**Published:** 2021-06-24

**Authors:** Tomoko Okuyama, Mayu Kyohara, Yasuo Terauchi, Jun Shirakawa

**Affiliations:** 1Department of Endocrinology and Metabolism, Graduate School of Medicine, Yokohama City University, Yokohama 236-0004, Japan; oku_tomo@yokohama-cu.ac.jp (T.O.); kyohara@yokohama-cu.ac.jp (M.K.); terauchi@yokohama-cu.ac.jp (Y.T.); 2Laboratory of Diabetes and Metabolic Disorders, Institute for Molecular and Cellular Regulation (IMCR), Gunma University, Maebashi 371-8510, Japan

**Keywords:** insulin-like growth factors, IGF-I receptor, insulin receptor, diabetes

## Abstract

It has been well established that insulin-like growth factors (IGFs) mainly mediate long-term actions in cell fates, whereas insulin predominantly exerts its role on metabolic activity. Indeed, insulin mediates multiple anabolic biological activities in glucose and amino acid transport, lipid and protein synthesis, the induction of glycogen, the inhibition of gluconeogenesis, lipolysis, and protein degradation. The interactions and differences between insulin receptor signaling and IGF-I receptor signaling in the metabolism and the cell fates are quite complicated. Because of the overlapping actions of IGF-I singling with insulin signaling, it has been difficult to distinguish the role of both signaling mechanisms on the metabolism. Furthermore, comprehensive information on the IGF-I function in respective tissues remains insufficient. Therefore, we need to clarify the precise roles of IGF-I signaling on the metabolism separate from those of insulin signaling. This review focuses on the metabolic roles of IGFs in the respective tissues, especially in terms of comparison with those of insulin, by overviewing the metabolic phenotypes of tissue-specific IGF-I and insulin receptor knockout mice, as well as those in mice treated with the dual insulin receptor/IGF-I receptor inhibitor OSI-906.

## 1. Introduction

Insulin-like growth factors (IGFs: IGF-I and IGF-II) have been shown to possess a variety of bioactivities in growth, cell differentiation, cell survival, and the maintenance of cell function. IGFs are expressed in multiple cell types. Most of the circulating IGF-I/II is secreted from hepatocytes [1,2]. IGF signaling is predominantly modulated by the growth hormone (GH), while multiple other factors, such as growth factors, tropic hormones, steroid hormones, erythropoietin, or the extracellular matrix, also regulate the IGF-I secretion from the liver [3].

The structure of insulin is well known to be highly homologous to those of IGFs [4]. Insulin and IGF-I act through their respective receptors. IGFs and insulin have been shown to share common signaling pathways, even though the physiological functions of insulin and IGFs are not identical. The IGF-I receptor (IGF-IR) is expressed highly in embryos and various tissues, except the liver and adipocytes of adults, whereas the insulin receptor (IR) is dominantly expressed in the liver, adipose tissue, and muscle [5]. The relative expression levels of IGF-IR and IR vary between tissues and across development stages, and these differences serve to dictate the relative tissue-specific role of IGF-I or insulin in regulating metabolic function. IGF-I is the principal ligand for IGF-IR and IR/IGF-IR hybrid receptors. IGF-I and insulin are able to activate each other’s receptors only at high concentrations. On the other hand, IGF-II can bind to IGF-IR as well as hybrid receptors and IR-A (a splicing variant of IR). Since IGFs potently bind to IGF-IR and activate pathways associated with cellular proliferation, the IGF axis has been recognized for its contribution to cancer growth and a propensity for metastasis. It is well known that cancer tissues express high levels of both IR and IGF-IR. However, in many common solid tumors, IGF axis deregulation is not itself the driver but occurs secondary to another molecular event that influences the expression of the ligands and/or receptors [6]. Since insulin receptor signaling also stimulates cell growth, hyperinsulinemia can potentially induce pathological disorders, including cancer and cardiovascular disease. Numerous in vitro and in vivo studies have now clearly established that insulin may affect the tumor progression by acting through its own receptor and not by crosstalk with IGF-IR [7].

IGF-IR is a disulfide bond-conjugated tetrameter that consists of two α subunits that are extracellular proteins binding to specific ligands and two β subunits that are transmembrane proteins possessing tyrosine kinase activity [8]. IGF-IR is highly homologous to IR, particularly in the cytoplasmic domain. Both splice variants of the IR (IR-A and IR-B) use signaling pathways that are similar to that of IGF-IR [9]. IGF-IR possesses a high affinity for IGF-I and IGF-II, but it can also bind to insulin with a lower affinity (Figure 1). In contrast, insulin receptor type B (IR-B), which is an alternative splicing variant of the insulin receptor gene, binds to insulin with high affinity and binds to IGFs with low affinity [9]. Another splicing variant of IR (IR-A) has a high affinity for both insulin and IGF-II. These differences in specificity play important roles in the distinct bioactivities of IGFs and insulin. It has also been shown that IGF-IR and IR can interact by forming hybrid receptors or by cross-talking in downstream intracellular signaling pathways, enabling them to modulate the amplitude of each signaling via the multiplicity of ligand-receptor interactions [10,11].

IGF-IR and IR are activated and auto-phosphorylated, followed by the binding of their ligands to the receptor. Subsequently, insulin receptor substrate (IRS)-1 and IRS-2 bind to the activated IGF-IR and IR through its phosphotyrosine binding (PTB) domain [12]. The activated IGF-IR or IR phosphorylates specific substrates, such as IRS-1, IRS-2, and Shc. The phosphorylated tyrosine residues of IRS-1, IRS-2, and Shc are recognized by various signaling molecules that contain an Src homology 2 (SH2) domain, including Grb2 and the 85 kDa regulatory subunit of phosphatidylinositol 3-kinase (PI3-kinase), leading to the activation of multiple downstream cascades (Figure 1). Grb2 is involved in the activation of a Ras-MAPK pathway. PI3-kinase, which consists of a regulatory subunit responsible for binding to IRSs and a catalytic subunit responsible for the phosphorylation of phosphatidylinositols, is a lipid kinase and a key element in the pathway leading to the metabolic effects of insulin. Thus, IGF-I has also been suggested to mediate insulin-like bioactivities.

To understand the roles of IGF-I/IGF-IR signaling in the regulation of metabolism and to distinguish the action of IGF-I/IGF-IR signaling from that of insulin/IR signaling, several reports have taken advantage of the Cre recombinase/loxP system to manipulate IGF-I and IGF-IR in a tissue-specific fashion (Table 1). In addition, we previously reported the role of IGF-IR and IR signaling in multiple metabolic organs, including the liver, adipose tissue, and pancreas, by using an orally bioavailable dual IGF-IR/IR inhibitor OSI-906 [13,14,15,16].

## 2. IGF-I and Insulin Signaling in Metabolic Organs

### 2.1. IGF-I and Insulin Signaling in the Liver

Insulin is directly secreted from pancreatic β-cells into the portal system, and the liver degrades about half of the insulin that reaches the liver [17]. Thus, the liver is the major target tissue of insulin action. IR- and IGF-IR-mediated signaling play a crucial role in hepatic insulin action, which regulates glucose and fatty acid metabolism. Yakar et al. generated liver-specific IGF-I-deficient (LID) mice and reported their metabolic phenotype [18]. The LID mice showed a 75% reduction in circulating IGF-I and elevated GH levels [18]. The LID mice also showed a 4-fold increase in the serum insulin levels and abnormal glucose clearance after insulin injection. In contrast, the fasting blood glucose levels and those after a glucose tolerance test were similar between LID mice and their control littermates. Thus, liver-specific IGF-I deficiency induced insulin resistance and marked hyperinsulinemia. Since liver-specific IGF-I deletion resulted in a marked reduction in the circulating total IGF-I levels and elevated GH levels, a reduction in insulin sensitivity was observed specifically in the muscle tissues of LID mice [18]. Thus, circulating IGF-I acted as a component of overall insulin action in peripheral tissues.

In contrast, hepatocyte-specific insulin receptor knockout (LIRKO) mice exhibited severe insulin resistance and glucose intolerance, leading to a significant increase in β-cell mass as well as the failure of insulin to suppress hepatic glucose production. Glycogen storage was also impaired in juvenile mice. LIRKO mice also showed an age-dependent progression of liver dysfunction without steatosis [19].

Collectively, these reports indicate the significance of IR and IGF-IR signaling in hepatic insulin action and the metabolism of glucose and fatty acids through inter-organ networks.

### 2.2. IGF-I and Insulin Signaling in Skeletal Muscle

Insulin has an anabolic effect on the muscle, controlling muscle protein synthesis and inhibiting muscle protein breakdown [20,21]. Because IR and IGF-IR are highly homologous receptors, IGF-I also mediates the regulation of skeletal muscle hypertrophy and atrophy. IGF-I increases the skeletal muscle protein synthesis via PI3K/Akt/mTOR and PI3K/Akt/GSK3β pathways and potentiates skeletal muscle regeneration via the activation of skeletal muscle stem (satellite) cells [22]. IGF-IR is expressed in undifferentiated myoblasts, whereas IR is predominantly expressed in differentiated muscles, compared with IGF-IR [23]. In addition, insulin activates glucose uptake into skeletal muscle, which contributes to the maintenance of glucose homeostasis. On the other hand, the effects of IGF-IR signaling in the muscle on glucose metabolism might be limited.

Laustsen et al. [24] generated muscle-specific IGF-IR knockout mice by crossing IGF- IR-floxed mice with muscle creatine kinase (MCK) promoter-Cre mice. Although MCK is expressed in the heart as well as the skeletal muscle, these mice had a comparable ratio of heart weight to body weight and cardiac performance. There were also no differences in body weight and glucose tolerance at the age of 6 months. O’Neill et al. [25] also generated mice with IGF-IR knockout restricted to the striated muscle by crossing IGF-IR-floxed mice with transgenic Cre mice under the human skeletal muscle actin promoter (ACTA1-Cre) [26], which is active in differentiated muscle cells [22]. These mice had a normal body weight and skeletal muscle mass up to the age of 52 weeks. Glucose intolerance was not observed even after high-fat-diet feeding, compared with the controls. Mavalli et al. [27] also generated mice with IGF-IR knockout in skeletal muscle by crossing IGF-IR-floxed mice with transgenic Cre mice under the control of the Mef2c skeletal muscle promoter. Mef2c is expressed in the skeletal muscle during development and postnatally [28]. These mice exhibited an abnormal muscle development characterized by a reduction in the myofiber number and area, and they failed to attain body weight and lean/muscle mass similar to that of the control mice after aging. However, no differences in serum glucose or triglycerides levels were seen between the two genotypes. These data suggested that IGF-IR signaling in skeletal muscle mainly contributes to muscle development and not to metabolic regulation.

Although there were some phenotypic differences depending on the Cre recombinase that was used, the absence of IR-induced peripheral adiposity and metabolic disorders, including glucose intolerance, was assumed to have been caused by the reduction in insulin-stimulated glucose uptake into the skeletal muscle. Laustsen et al. [24] and Brüning et al. [29] generated muscle-specific IR knockout mice by crossing IR-floxed mice with MCK-Cre mice. These mice were comparable for body weight, whereas had a decreased ratio of heart weight to body weight and tended to have impaired cardiac performance. They also had an increased fat mass and elevated serum triglyceride and free fatty acids levels, whereas their glucose tolerance was normal. O’Neill et al. [25] also generated striated muscle-specific IR knockout mice by crossing ACTA1-Cre mice [26] with IR-floxed mice. There were no differences in body weight and skeletal muscle mass up to 52 weeks of age. A high-fat diet did not induce glucose intolerance in these mice. Additionally, mice overexpressing IR specifically in skeletal muscles had an increased lean mass, were protected from high-fat-diet-induced obesity, and had an improved glucose tolerance, all of which were associated with the reduction of inflammation in fat and liver tissue [30]. On the other hand, insulin-stimulated Akt phosphorylation was reduced in the muscles of those mice. It was suggested that the chronic overstimulation of IR led to post-receptor insulin resistance in muscle, indicating that a balance in the activity of IR signaling is critical.

Furthermore, Laustsen et al. [24] created the muscle-specific IR and IGF-IR double knockout mice using MCK promoter in transgenic mice, which had a reduced heart weight and developed heart failure, whereas had normal blood glucose and insulin levels. O’Neill et al. [25] also created the muscle-specific IR and IGF-IR double knockout mice using ACTA1 promoter in transgenic mice, which exhibited severe muscle atrophy but normal glucose tolerance.

In summary, IR/IGF-IR signaling in muscle plays an important role in muscle growth and glucose metabolism. In particular, IGF-IR mainly regulates muscle development.

### 2.3. IGF-I and Insulin Signaling in Adipose Tissue

Insulin and IGF-I action through their respective receptors are essential for the development and function of brown adipose tissue (BAT) and white adipose tissue (WAT) [31,32]. To investigate the direct impact of IGF-I on adipocytes, Kloting et al. crossed aP2-Cre mice with IGF-IR floxed mice [33]. Although these mice had an increased fat mass and enlarged adipocyte size, the recombination of the floxed alleles was not specific to the adipose tissue as it was also observed in the peripheral nervous system of aP2 promoter mice. Subsequently, adipose-tissue-specific IGF-IR knockout mice were established by crossbreeding adiponectin-Cre mice with IGF-IR floxed mice [34]. These adipocyte-specific IGF-IR knockout mice (F-IGFRKO) showed a reduction in fat pad mass (~25%) both in subcutaneous and brown adipose tissue, decreased lipogenic gene expression, and lower plasma leptin and adiponectin levels. In contrast, glucose tolerance and insulin sensitivity were not influenced by the adipocyte-specific IGF-IR deletion.

Mice with adipocyte-specific knockout for both insulin and IGF-I receptors (FIGIRKO) had markedly decreased white and brown fat mass [32]. FIGIRKO mice were unable to maintain their body temperature when placed in an environment at 4 °C. However, basal energy expenditure was increased in the FIGIRKO mice, despite a more than 85% reduction in brown fat mass. These FIGIRKO mice were also protected against high-fat-diet-induced obesity and age- and high-fat-diet-induced glucose intolerance. The brown fat activity was markedly decreased in the FIGIRKO mice but was responsive to β3-receptor stimulation. Thus, insulin/IGF-I signaling has a crucial role in the control of brown and white adipose tissue development, and a deficiency of both signaling pathways leads to abnormal thermogenesis and a paradoxical increase in the basal metabolic rate. In contrast, mice with a fat-specific disruption of the insulin receptor gene (FIRKO mice) had a 95% reduction in white adipose tissue but a paradoxical 50% increase in brown adipose tissue with the accumulation of large unilobular lipid droplets [34]. Consistent with a reduction in fat mass, the plasma leptin levels were decreased in FIRKO mice. In addition, FIRKO mice were protected against age-related and hypothalamic lesion-induced obesity and obesity-related glucose intolerance. FIRKO mice also exhibited a polarization of adipocytes into populations of large and small cells, which differed in the expression of fatty acid synthase, C/EBP alpha, and SREBP-1. Both FIGIRKO and FIRKO mice were unable to maintain a body temperature under cold exposure, and they exhibited ectopic lipid accumulation in the liver and muscle as well as pancreatic islet hyperplasia. FIRKO and FIGIRKO mice also developed overt diabetes, severe insulin resistance, and significant dyslipidemia.

Thus, insulin signaling in adipocytes is critical for the development of obesity and its associated metabolic abnormalities, and the abrogation of insulin signaling in fat unmasks a heterogeneity in the adipocyte response in terms of gene expression and triglyceride storage. Taken together, insulin and IGF-I action through IR and IGF-IR coordinately play a crucial role in adipose tissue development, its function, energy metabolism, and glucose metabolism.

### 2.4. IGF-I and Insulin Signaling in Pancreatic β-Cells

IGF-IR/IR signaling in β-cells is essential for their adaptive proliferation and the maintenance of their function as insulin secretory cells. Impaired IGF-IR/IR signaling in β-cells contributes to a decrease in insulin secretion and a mass reduction in β-cells.

Kulkarni et al. [35] generated β-cell-specific IGF-IR knockout (βKO) mice by crossing IGF-IR-floxed mice with β-cell-specific rat-insulin-2 promoter (RIP)-Cre mice. The βIGF-IRKO mice showed normal growth and β-cell development and did not have an altered β-cell mass, but they had a reduced expression of GLUT2 and glucokinase in β-cells, resulting in defective glucose-stimulated insulin secretion and impaired glucose tolerance. Xuan et al. [36] also generated βIGF-IRKO mice using different strains of RIP-Cre mice; these mice exhibited defective glucose- and arginine-dependent insulin secretion and impaired glucose tolerance. However, no difference in β-cell mass or proliferation was seen. Thus, IGF-I in β-cells is crucial for insulin secretion but not for the regulation of β-cell proliferation.

IGF-II, as well as IGF-I and insulin, is a potent ligand of IGF-IR. IGF-I is expressed at a low level or is not detectable, whereas IGF-II is highly expressed in the β-cells of adult mice [37] and humans [38]. Honey Modi et al. [39] generated β-cell-specific IGF-II knockout (βIGFIIKO) mice by crossing IGF-II-floxed mice with Ins-1-Cre mice. Inoel Sandovici et al. [40] also generated βIGFIIKO mice by crossing IGF-II-floxed mice with RIP-Cre mice. They suggested that the autocrine actions of IGF-II impacted β-cell plasticity under conditions of increased metabolic demand, especially in females. Several reports have indicated that the overexpression of IGF-II increases β-cell expansion and replication in rodents [41,42,43]. In addition, Marion Cornu et al. [37] revealed that GLP-1 stimulated IGF-IR expression and IGF-IR signaling via autocrine IGF-II in β-cells, leading to the protection of β-cells from apoptosis. On the other hand, Yarong Lu et al. [44] generated mice with pancreatic-specific IGF-I gene deficiency (PID) by crossing Pdx1-Cre mice with IGF-I-floxed mice. The PID mice had an enlarged islet cell mass and were resistant to streptozotocin-induced diabetes. Thus, locally produced IGF-I within the pancreas might inhibit islet cell growth. Furthermore, IGFBPs (insulin-like growth factor binding proteins), which comprise a family of six proteins (1–6), bind to IGF-I and IGF-II in extracellular fluid. Jing Lu et al. [45] revealed that IGFBP-1 increased β-cell regeneration, at least in part, by inhibiting IGF-IR signaling. In addition, Nuria Palau et al. [46] revealed that the blockage of IGFBP-3 led to an increase in β-cell proliferation during obesity, probably through the suppression of IGF-IR signaling.

Kulkarni et al. [47] generated β-cell-specific IR knockout (βIRKO) mice by crossing IR-floxed mice with RIP-Cre mice. The βIRKO mice were born normally and showed no difference in islet morphology, but they exhibited impaired glucose tolerance, with decreased glucose-stimulated insulin secretion and a progressive reduction in the β-cell mass. Okada et al. [48] revealed that βIRKO mice failed to exhibit compensatory islet proliferation and hyperplasia in response to high-fat-diet feeding. Otani et al. [49] reported that a modest reduction in the glucose-sensing associated gene, GLUT2, and glucokinase gene expression in the islets of βIRKO mice, resulting in defective glucose-stimulated insulin secretion and impaired glucose tolerance in βIRKO mice. In addition, Shirakawa et al. [50] revealed that IR signaling promoted adaptive β-cell proliferation in response to pregnancy, acute/chronic insulin resistance, and aging through the FoxM1/PLK1 (polo-like kinase-1)/CENP-A (centromere protein A) pathway-mediated mitotic cell-cycle progression.

In addition, Ueki et al. [51] created the β-cell-specific IR and IGF-IR double knockout (βDKO) mice by breeding the RIP-Cre transgenic βIRKO mice and βIGF-IRKO mice. The βDKO mice were born normally, and no differences in β-cell mass or islet architectures were seen between the βDKO mice and their controls at birth. However, the βDKO mice had a reduced β-cell mass at 2 weeks of age and developed overt diabetes at 3 weeks of age, representing a worse diabetic phenotype than that seen in single knockout models of IR or IGF-IR in β-cells. βIR−/−βIGF-IR+/− mice exhibited severe hyperglycemia with a reduced β-cell mass at 2 weeks of age, whereas βIR+/−βIGF-IR−/− mice exhibited mild hyperglycemia with no change in the β-cell mass. These results suggested that IR participated in the maintenance of β-cell mass rather than IGF-IR.

Recently, Ansarullah et al. [52] reported an insulin inhibitory receptor, inceptor. Inceptor calibrates IR and IGF-IR internalization via clathrin-mediated endocytosis, thereby regulating the ligand sensitization of IR and IGF-IR in β-cells. β-cell-specific inceptor knockout mice exhibited an increase in IR and IGF-IR signaling, enhanced β-cell proliferation, and improved glucose tolerance. Furthermore, treatment with an anti-inceptor neutralizing monoclonal antibody resulted in the sustained activation of IR and IGF-IR in β-cells. The expression of IR-mediated signaling molecules is known to be reduced in islets from type 2 diabetes patients [53]. Thus, treatments targeted at IR/IGF-IR signaling via inceptor might contribute to the development of radical treatments to recover β-cell function in type 2 diabetes patients.

In addition, whether IR/IGF-IR itself is needed in β-cells remains controversial. Several reports of mice deficient for molecules involved in the IR/IGF-IR-mediated signaling pathway in β-cells have been made. Kubota et al. [54] generated β-cell-specific IRS-2 knockout mice by crossing IRS-2-floxed mice with RIP-Cre mice, which exhibited impaired glucose-induced insulin secretion and reduced β-cell proliferation. Hashimoto et al. [55] generated β-cell-specific PDK-1 knockout mice by crossing PDK-1-floxed mice with RIP-Cre mice, which reduced the β-cell mass and resulted in hyperglycemia with a reduction in plasma insulin levels. Thus, the downstream signaling of IR/IGF-IR is obviously critical for β-cell maintenance, whereas it is unclear whether the interaction of ligands with IR/IGF-IR is required for β-cell function. Since IR/IGF-IR-mediated signaling molecules, such as IRS proteins, form complexes by binding to IR/IGF-IR, those receptors might act as anchors for signaling complexes in a manner that is independent of their ligands.

In summary, IGF-IR signaling in β-cells contributed to β-cell functions and β-cell proliferation. IR and IGF-IR signaling compensate for each other’s actions in β-cells.

### 2.5. IGF-I and Insulin Signaling in the Brain

IR and IGF-IR are also expressed in the central nervous system. Those two receptors have different physiological functions in the brain, as shown by genetic knockout in mice. Mice with the genetic deletion of IGF-IR in the brain had a reduced brain size, generalized growth retardation, and behavioral changes. In contrast, mice with the genetic deletion of IR in the brain (neuron-specific IR knockout; NIRKO) had a normal brain size and development, but they exhibited metabolic phenotypes, including mild obesity and insulin resistance [56]. In fact, female NIRKO mice showed an increased food intake, and both male and female mice developed diet-sensitive obesity with increases in body fat and plasma leptin levels, mild insulin resistance, elevated plasma insulin levels, and hypertriglyceridemia. NIRKO mice also exhibited impaired spermatogenesis and ovarian follicle maturation because of the hypothalamic dysregulation of the luteinizing hormone. Thus, IR signaling in the CNS plays an important role in the regulation of energy disposal, fuel metabolism, and reproduction. Interestingly, the heterozygous deletion of IGF-IR in the brain did not affect the brain growth, but it resulted in a complex phenotype that included reduced body weight, increased fat mass, impaired glucose tolerance, and an extended lifespan [57].

## 3. Metabolic Phenotype in OSI-906 Treated Mice

OSI-906 (linsitinib), an anti-tumor drug, specifically inhibits the autophosphorylation of both IR and IGF-IR, resulting in the induction of insulin resistance [58]. OSI-906 has been shown to block the ligand-induced activation of downstream pathways, including the phosphorylation of Akt, Erk1/2, and p70S6K, in the 3T3/huIGF-IR cell line (LISN) [13]. The inhibition of IR/IGF-IR signaling by OSI-906 in a two-stage BALB/c-3T3 cell transformation assay (BALB-CTA) diminished the phosphorylation of Akt, p70S6K, S6 protein, 4EBP-1, GSK3β, and AMPK, but not the phosphorylation of Erk1/2 [14]. The daily administration of OSI-906 to mice for a week induced increased β-cell mass and β-cell proliferation, accompanied by hyperinsulinemia, glucose intolerance, lipodystrophy, and liver steatosis [13,14] (Figure 2).

We previously reported that the systemic inhibition of IR and IGF-IR with OSI-906 induced acute insulin resistance in mice [13]. The oral administration of OSI-906 evoked glucose intolerance in a dose-dependent manner in wild-type mice. The serum glucose and insulin levels were immediately elevated an hour after OSI-906 administration, and continuous hyperglycemia and hyperinsulinemia were observed during the treatment with OSI-906 until day 7. The mice treated with OSI-906 for a week showed a significantly increased β-cell mass and β-cell proliferation. These results suggested that β-cell proliferation was mediated by an insulin signaling-independent pathway, as no change in the cyclin D2 expression was observed in the mouse islets [14]. Notably, the withdrawal of OSI-906 after a week of administration reversed the hyperglycemia and hyperinsulinemia, as well as the β-cell proliferation, although the increased β-cell mass was restored [14].

Mice treated with OSI-906 for 7 days showed lipoatrophy in BAT and WAT with elevated serum free fatty acid levels (Figure 2). The expression of leptin, but not adiponectin, in fat was significantly lower in the OSI-906-treated mice, compared with that in control mice [14]. Conditional IGF-IR inactivation was previously shown to result in an increased WAT mass and did not affect the BAT mass or its activity [33]. These findings imply differences in the roles of IGF-IR and IR signaling on the regulation of adipose tissue mass.

Insulin signaling in the liver was impaired in mice treated with OSI-906, resulting in elevated gluconeogenic gene expressions. Mice treated with OSI-906 for 7 days showed hepatic steatosis (Figure 2). The liver weight, as well as both liver triglyceride and liver glycogen contents, were significantly increased by the administration of OSI-906. On the other hand, the gene expressions involved in lipogenesis, such as Srebp1c, Fas, and Scd1, were not altered by treatment with OSI-906 in the liver [14,16]. Notably, leptin supplementation on OSI-906-treated mice reduced hepatic triglyceride accumulation and improved steatosis without improving lipodystrophy. Interestingly, liver steatosis was ameliorated after the withdrawal of OSI-906, which coincided with the recovery from lipodystrophy [14] (Figure 2). These findings indicate the inter-organ network between the liver and adipose tissues in the onset or remission of IR and IGF-IR signaling-mediated hepatic steatosis.

## 4. Perspective

In this review, we focused on the phenotypes of conditional knockout models of the IR/IGF-IR and outlined the roles of the IGF-I axis on glucose metabolism in metabolic organs. A defect in IGF-IR signaling contributes to the development of diabetes. Thus, anti-diabetic drugs that target the IGF-I axis have the potential to be useful for human diabetes patients. On the other hand, the activation of IGF-IR signaling is generally considered to promote carcinogenesis. Diabetes patients are at a high risk of cancer because of the increase in IGF-IR activities arising from compensatory hyperinsulinemia for insulin resistance. Metformin, one of the most broadly used agents for the treatment of type 2 diabetes patients, has been shown to have anti-cancer effects because of its suppression of IGF-I/IGF-IR signaling activities by multiple mechanisms, such as the activation of AMP-activated protein kinase (AMPK) and the associated inhibition of mTOR (mechanistic target of rapamycin) signaling [59]. Metformin has been reported to have a tumor suppressor effect in many cancer types in large-scale clinical studies involving many human type 2 diabetes patients, in addition to analyses using cell lines or animal models. The development of drugs that have both anti-cancer and anti-diabetic effects to modulate IGF-I/IGF-IR signaling activity appropriately is expected in the future.

The elucidation of the role of the IGF-I axis in metabolism will also contribute to the development of the field of aging. IGF-IR heterozygous knockout mice live, on average, 26% longer than their wild-type littermates and display greater resistance to oxidative stress, a known determinant of aging [60]. In addition, the inhibition of the TOR (target of rapamycin) signaling pathway, which is a nutrient response signaling pathway, extends the lifespan of Caenorhabditis elegans [61] and Drosophila [62]. Caloric restriction is known to extend the lifespan in animal models, which might be involved in IGF-IR signaling through the suppression of mTOR signaling. Furthermore, the increase in systemic low-grade inflammation in the elderly is referred to as “inflammaging,” which contributes to the development of chronic diseases, including diabetes, and accelerates aging. GH receptor-deficient mice reduced the circulating levels of IGF-I, resulting in a prolonged lifespan, and protected them from age-related inflammasome activation and associated T cell senescence [63]. Additionally, in humans, the presence of heterozygous mutations in the IGF-IR gene [64] or GH1 (human growth hormone) gene SNPs (single nucleotide polymorphisms) [65] reportedly affect longevity, indicating an involvement between IGF-IR signaling and lifespan.

In this way, the IGF-I axis is a principal mechanism that links diabetes, cancer, and aging. If a drug that modulates the IGF-I axis is developed in the future, it could become a silver bullet that solves these clinically important issues simultaneously. Further elucidation of the roles of the IGF-I axis from various perspectives is expected.

## Figures and Tables

**Figure 1 ijms-22-06817-f001:**
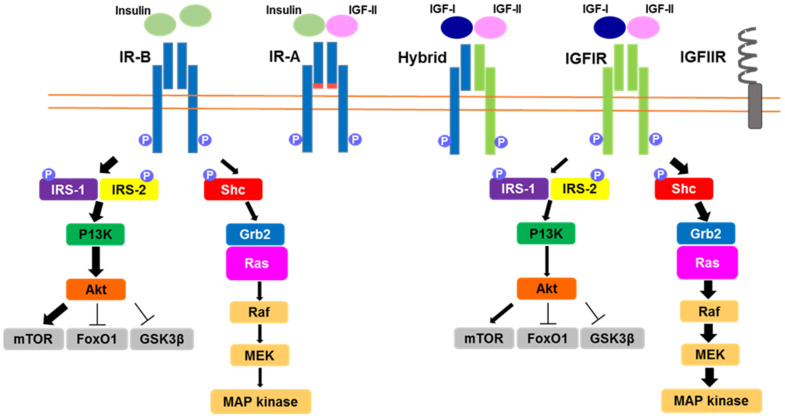
Schematic illustration of IGF-I receptor/insulin receptor and downstream intracellular signaling. The IGF-I receptor (IGF-IR) and the insulin receptor (IR: IR-A and IR-B) possess a respective affinity for insulin, IGF-I, and IGF-II. IGF-IR and IR can interact by forming hybrid receptors or by cross-talking in downstream intracellular signaling pathways. The sizes of the arrows show the amplitudes of each signaling pathway within the multiplicity of ligand-receptor interactions.

**Figure 2 ijms-22-06817-f002:**
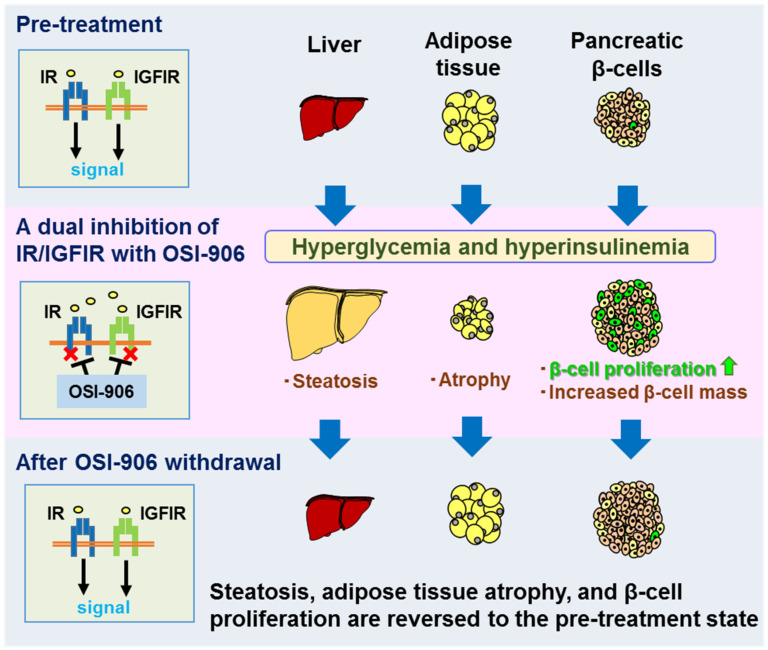
The effect of administration and withdrawal of OSI-906 on metabolic organs. The daily administration of OSI-906, a dual inhibitor of IGF-I receptor (IGF-IR) and insulin receptor (IR), to mice for a week provokes hepatic steatosis, induces lipoatrophy, and increases β-cell proliferation accompanied by hyperglycemia and hyperinsulinemia. The withdrawal of OSI-906 reverse hepatic steatosis, lipoatrophy, and β-cell proliferation, although the increased β-cell mass remained.

**Table 1 ijms-22-06817-t001:** The phenotype of the tissue-specific IGF-I receptor, insulin receptor, or insulin receptor/IGF-I receptor double-knockout mice.

Tissue	Promoter-Driven Cre	IGF-IRKO	IRKO	DKO
Liver	Albumin	-	severe insulin resistance	-
(Hepatocyte)			overt severe diabetes	
			increase in β-cell mass	
			liver dysfunction (age-related)	
			reduced serum triglycerides and free fatty acids	
Muscle	Creatine kinase	normal body weight	normal body weight	reduced body weight
		normal glucose tolerance	normal glucose tolerance	normal glucose tolerance
		normal cardiac performance	impaired cardiac performance	developed heart failure
			elevated serum triglycerides and free fatty acids	
	ACTA1	normal body weight	normal body weight	severe muscle atrophy
		no change in muscle mass	no change in muscle mass	normal glucose tolerance
		normal glucose tolerance	normal glucose tolerance	
	Mef2c	normal body weight	-	-
		normal serum glucose and triglycerides		
Adipose tissue	Adiponectin	reduced WAT and BAT mass	reduced body weight	reduced body weight
		decreased lipogenic gene expression	reduced WAT and increased BAT mass	reduced WAT and BAT mass
		lower plasma leptin and adiponectin level	lower plasma leptin and adiponectin level	lower plasma leptin and adiponectin level
		normal glucose tolerance	overt diabetes	overt diabetes
		normal insulin tolerance	severe insulin resistance	severe insulin resistance
			ectopic lipid accumulation	ectopic lipid accumulation
			dyslipidemia	dyslipidemia
			pancreatic islet hyperplasia	pancreatic islet hyperplasia
			cold intolerance	severe cold intolerance
				increased basal energy expenditure
Pancreatic β-cell	Rat insulin 2 promoter	no change in β-cell mass and β-cell proliferation	reduced β-cell mass and β-cell proliferation	markedly reduced β-cell mass
		impaired glucose- and arginine-induced insulin secretion and impaired glucose tolerance	impaired glucose-stimulated insulin secretion and impaired glucose tolerance	overt diabetes
Brain	Rat nestin promoter	reduced brain size	normal brain size	
		growth retardation	normal growth	
		behavioral changes	mild obesity	
			insulin resistance	

IGF-IRKO, IGF-I receptor knockout; IRKO, insulin receptor knockout; DKO, IGF-I receptor/insulin receptor double knockout; BAT, brown adipose tissue; WAT, white adipose tissue.

## Data Availability

Not applicable.

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
