# Peer review of "The Roles of the IGF Axis in the Regulation of the Metabolism: Interaction and Difference between Insulin Receptor Signaling and IGF-I Receptor Signaling"

_ijms, 2021, doi:10.3390/ijms22136817_

Round 1
Reviewer 1 Report
They reviewed the roles of IGF system in the regulation of metabolism.
Their text is good for understanding of IGF axis in metabolism.
If they showed more figures, the readers will understand this mechanism well.
Author Response
Reviewer #1
We are grateful to Reviewer #1 for extensive review of the previous version of this manuscript and making suggestion which is helpful in the revision of the manuscript.
Responses to Comments
Comment:
Their text is good for understanding of IGF axis in metabolism. If they showed more figures, the readers will understand this mechanism well.
Response:
We thank the Reviewer#1 for this suggestion. We now included an additional figure, in association with the effect of a dual IR/IGF1R inhibitor OSI-906 on metabolic phenotype, as Figure 2 in the manuscript.

Reviewer 2 Report
This is an apropriate and interesting summary of the progress of research made on the overlap of two major signal pathways. The conclusions are moderate and forward-looking. The work itself will be helpful for educational reason and to comprehend major obstacles of the field.
Author Response
Reviewer #2
We are grateful to Reviewer #2 for careful and extensive review of the previous version of this manuscript.
Responses to Comments
Comment:
This is an appropriate and interesting summary of the progress of research made on the overlap of two major signal pathways. The conclusions are moderate and forward-looking. The work itself will be helpful for educational reason and to comprehend major obstacles of the field.
Response:
We appreciate the Reviewer#2 for this comment.
